# Efficient Identification in Linear Structural Causal Models with Instrumental Cutsets

**Daniel Kumor**
Purdue University
dkumor@purdue.edu

**Bryant Chen**
Brex Inc.
bryant@brex.com

**Elias Bareinboim**
Columbia University
eb@cs.columbia.edu

## Abstract

One of the most common mistakes made when performing data analysis is attributing causal meaning to regression coefficients. Formally, a causal effect can only be computed if it is identifiable from a combination of observational data and structural knowledge about the domain under investigation (Pearl, 2000, Ch. 5). Building on the literature of instrumental variables (IVs), a plethora of methods has been developed to identify causal effects in linear systems. Almost invariably, however, the most powerful such methods rely on exponential-time procedures. In this paper, we investigate graphical conditions to allow efficient identification in arbitrary linear structural causal models (SCMs). In particular, we develop a method to efficiently find unconditioned *instrumental subsets*, which are generalizations of IVs that can be used to tame the complexity of many canonical algorithms found in the literature. Further, we prove that determining whether an effect can be identified with TSID (Weihs et al., 2017), a method more powerful than unconditioned instrumental sets and other efficient identification algorithms, is NP-Complete. Finally, building on the idea of flow constraints, we introduce a new and efficient criterion called *Instrumental Cutsets* (IC), which is able to solve for parameters missed by all other existing polynomial-time algorithms.

## 1 Introduction

Predicting the effects of interventions is one of the fundamental tasks in the empirical sciences. Controlled experimentation is considered the "gold standard" in which one physically intervenes in a system to learn about the corresponding effects. In practice, however, experimentation is not always possible due costs, ethical constraints, or technical feasibility – e.g., a self-driving car should not need to crash to recognize that doing so has negative consequences. In such cases, an agent must determine the effect of an action using observational data and knowledge of the environment's structure. This is known as the problem of *identification* (Pearl, 2000; Bareinboim & Pearl, 2016).

Structural knowledge is usually represented as a structural causal model (SCM)[1] (Pearl, 2000), which represents the set of observed and unobserved variables and their corresponding causal relations as a system of equations. We focus on the problem of generic identification in linear, acyclic SCMs. In such systems, the value of each observed variable is determined by a linear combination of the values of its direct causes along with a latent error term $\epsilon$. This leads to a system of equations $X = \Lambda^T X + \epsilon$, where $X$ is the vector of variables, $\Lambda^T$ is a lower triangular matrix whose $ij$th element $\lambda_{ij}$ – called *structural parameter* – is 0 whenever $x_i$ is not a direct cause of $x_j$, and $\epsilon$ is a vector of latent variables.

Methods for identification in linear SCMs generally assume that variables are normally distributed (Chen & Pearl, 2014), meaning that the observational data can be summarized with a covariance matrix $\Sigma$. This covariance matrix can be linked to the underlying structural parameters through the

system of equations $\Sigma = XX^T = (I - \Lambda)^{-T}\Omega(I - \Lambda)^{-1}$ (1), where $\Omega$'s elements, $\epsilon_{ij}$, represent $\sigma_{\epsilon_i \epsilon_j}$, and $I$ is the identity matrix (Foygel et al., 2012). The task of causal effect identification in linear SCMs can, therefore, be seen as solving for the target structural parameter $\lambda_{ij}$ using Eq. (1). If the parameter can be uniquely solved using $\Sigma$ alone, then it is said to be *generically identifiable*.

Such systems of polynomial equations can be approached directly through the application of Gröbner bases (García-Puente et al., 2010). In practice, however, these methods are doubly-exponential (Bardet, 2002) in the number of structural parameters, and become computationally intractable very quickly, incapable of handling causal graphs with more than 4 or 5 nodes (Foygel et al., 2012).

Identification in linear SCMs has been a topic of great interest for nearly a century (Wright, 1921), including much of the early work in econometrics (Wright, 1928; Fisher, 1966; Bowden & Turkington, 1984; Bekker et al., 1994). The computational aspects of the problem, however, have only more recently received attention from computer scientists and statisticians (Pearl, 2000, Ch. 5).

Since then, there has been a growing body of literature developing successively more sophisticated methods with increasingly stronger identification power i.e., capable of covering a larger spectrum of identifiable effects. Deciding whether a certain structural parameter can be identified in polynomial time is currently an open problem.

The most popular identification method found in the literature today is known as the *instrumental variable* (IV) (Wright, 1928). A number of extensions of IVs have been proposed, including *conditional IV* (cIV), (Bowden & Turkington, 1984; Van der Zander et al., 2015), unconditioned *instrumental sets* (IS) (Brito, 2004), and the *half-trek criterion* (HTC) (Foygel et al., 2012), all of which are accompanied with efficient, polynomial-time algorithms.

In contrast, generalized instrumental sets (gIS) (Brito & Pearl, 2002) were developed as a graphical criterion, without an efficient algorithm. Van der Zander & Liśkiewicz (2016) proved that finding conditioning sets that satisfy the gIS given a set of instruments is NP-Hard. They further proposed a simplified version of the criterion (scIS), for which finding a conditioning set can be done efficiently. It remains an open problem whether instruments satisfying the gIS criterion can be found in polynomial time.

The generalized HTC (gHTC) (Chen, 2016; Weihs et al., 2017) and auxiliary variables (AVS) (Chen et al., 2016, 2017) were developed with algorithms that were polynomial, provided that the number of incoming edges to each

### Identification Power & Efficiency

| Algorithm | Power | Eff.? |
|---|---|---|
| IV[a] | low | ✓ |
| cIV[b] | medium | ✓[f] |
| IS[c] | medium | ✓ |
| scIS[g] | high | $? \rightarrow$ ✗[*] |
| gIS[c] | high | ?[†] |
| HTC[e] | high | ✓ |
| gHTC[h,j] | very high | $? \rightarrow$ ✓[‡] |
| cAV & AVS[i] | very high | $? \rightarrow$ ✓[‡] |
| **Our Method** | **very high** | ✓ |
| gAVS[i] | very high | ?[†] |
| tsIV & gHTC[j] | very high | $? \rightarrow$ ✗[*] |
| Gröbner[d] | complete | ✗ |

[a]Wright (1928); [b]Bowden & Turkington (1984); [c]Brito & Pearl (2002); [d]García-Puente et al. (2010); [e]Foygel et al. (2012); [f]Van der Zander et al. (2015); [g]Van der Zander & Liśkiewicz (2016); [h]Chen (2016); [i]Chen et al. (2017); [j]Weihs et al. (2017).
[†] Finding conditioning set for candidates shown to be NP-hard by (g), but complexity of search is open question.
[‡] Previous algorithms exponential without bounded input degree.
[*] We proved NP-Completeness of this method.

Table 1: Our contributions in relation to the literature are shown in **red**; Ordered roughly by identification power. $? \rightarrow$ represents methods for which we determined complexity in this work.

node in the causal graph were bounded. The corresponding algorithms are exponential without this restriction, since they enumerate all subsets of each node's incident edges.

More recently, Weihs et al. (2017) showed how constraints stemming from determinants of minors in the covariance matrix (Sullivant et al., 2010) can be exploited for identification (TSID). Still, the complexity of their method was left as an open problem. We use the term *tsIV* to refer to the unnamed criterion underlying the TSID algorithm. Against this background, our contributions are as follows:

- We develop an efficient algorithm for finding instrumental subsets, which overcomes the need for enumerating all subsets of incoming edges into each node. This leads to efficient identification algorithms exploiting the gHTC and AVS criteria.

- We prove NP-Completeness of finding tsIVs and scIS, which shows they are impractical for use in large graphs without constraining the search space.

- Finally, we introduce a new criterion called *Instrumental Cutsets*, and develop an associated polynomial-time identification algorithm. We show that ICs subsume both gHTC and AVS.

For clarity, a summary of our results in the context of existing literature is shown in Table 1.

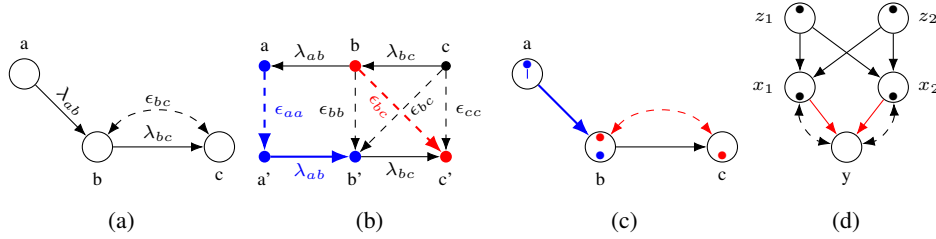

Figure 1: Conversion of an instrumental variable (a) into a trek-encoding flow graph is shown in (b) if one ignores the colorings. From it, we can deduce that $\sigma_{bc} = \lambda_{ab}\epsilon_{aa}\lambda_{ab}\lambda_{bc} + \epsilon_{bb}\lambda_{bc} + \epsilon_{bc}$. (c) shows another way of drawing the sets from (b), which facilitates interpretation in more complex settings. In (d), $z_1, z_2$ can be used as an instrumental set to solve for $\lambda_{x_1 y}$.

## 2 Preliminaries

The causal graph of an SCM is defined as a triple $G = (V, D, B)$, where $V$ represents the nodes, $D$ the directed edges, and $B$ the bidirected ones. A linear SCM's graph has a node $v_i$ for each variable $x_i$, a directed edge between $v_i$ and $v_j$ for each non-zero $\lambda_{ij}$, and a bidirected edge between $v_i$ and $v_j$ for each non-zero $\epsilon_{ij}$ (Fig. 1a). Each edge in the graph, therefore, corresponds to an unknown coefficient, which we call a *structural parameter*. When clear from the context, we will use $\lambda_{ij}$ and $\epsilon_{ij}$ to refer to the corresponding directed and bidirected edges in the graph. We define $Pa(x_i)$ as the set of parents of $x_i$, $An(x_i)$ as ancestors of $x_i$, $De(x_i)$ as descendants of $x_i$, and $Sib(x_i)$ as variables connected to $x_i$ with bidirected edges (i.e., variables with latent common causes).

We will refer to paths in the graph as "unblocked" conditioned on a set $W$ (which may be empty), if they contain a collider ($a \to b \leftarrow c, a \leftrightarrow b \leftrightarrow c, a \to b \leftrightarrow c$) only when $b \in W \cup An(W)$, and if they do not otherwise contain vertices from $W$ (see d-separation, Koller & Friedman (2009)). Unblocked paths without conditioning do not contain colliders, and are referred to as treks (Sullivant et al., 2010). The computable covariances of observable variables and the unknown structural parameters given in Eq. (1) have a graphical interpretation in terms of a sum over all treks between nodes in the causal graph, namely $\sigma_{xy} = \sum \pi(x, y)$, where $\pi$ is the product of structural parameters along the trek.

Since unblocked paths in the causal graph have a non-trivial relation to arrow directions, we follow Foygel et al. (2012) in constructing an alternate DAG, called the "flow graph" ($G_{flow}$), which encodes treks as directed paths between nodes (see Fig. 1b, where the blue path shows a trek between A and B in Fig. 1a, meaning $\sigma_{ab} = \epsilon_{aa}\lambda_{ab}$). When referencing the flow graph, we call the "top" nodes (e.g., $a, b, c$ in Fig. 1b) the "source nodes", and the "bottom" nodes ($a', b', c'$) the "sink nodes".

Treks between two sets of nodes in $G$ are said to have "no sided intersection" if they do not intersect in $G_{flow}$. The red and blue paths of Fig. 1b show such a set from $\{a, b\}$ to $\{b', c'\}$. Non-intersecting path sets are related to minors of the covariance matrix, denoted with $\Sigma_{(a,b),(b,c)}$[2]. We visually denote the source and sink sets in the original graph with a dot near the top of the node if it is a source, and a dot near the bottom if it is a sink. By these conventions, Fig. 1b can be represented by Fig. 1c.

For simplicity, we will demonstrate many of our contributions in the context of instrumental sets:

**Definition 2.1.** *(Brito & Pearl, 2002) A set $Z$ is called an **instrumental set (IS)** relative to $X \subseteq Pa(y)$ if (i) there exists an unblocked path set without sided intersection between $Z$ and $X$, and (ii) there does not exist an unblocked path from any $z \in Z$ to $y$ in $G$ with edges $X \to y$ removed.*

In Fig. 1d, $\{z_1, z_2\}$ is an instrumental set relative to $\{x_1, x_2\}$, leading to a system of equations solvable for $\lambda_{x_1 y}, \lambda_{x_2 y}$. A conditioning set $W$ can be added to block paths from $Z$ to $y$, creating the *simple conditional IS* (scIS). If each $z_i$ has its own conditioning set, it is called a *generalized IS* (gIS).

A set of identified structural parameters $\Lambda^*$ can be used to create "auxiliary variables" (Chen et al., 2016) which subtract out parents of variables whose effect is known: $x_i^* = x_i - \sum_{\lambda_{x_j x_i} \in \Lambda^*} \lambda_{x_j x_i} x_j$.

Using these variables as instruments leads to AV sets (AVS), which are equivalent to the gHTC[3]. Finally, we will build upon the tsIV, which exploits flow constraints in $G_{flow}$ to identify parameters:

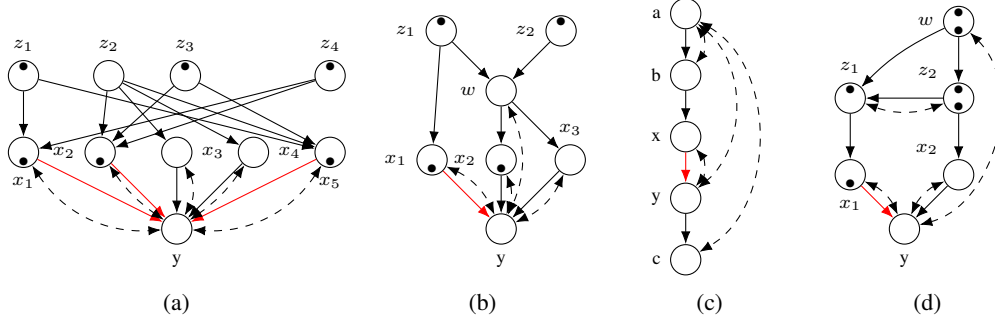

Figure 2: (a): only a subset of edges incident to $y$ can be identified with an instrumental set, and finding the maximal subset in arbitrary graphs was exponential in previous algorithms. (b): $\lambda_{x_1 y}$ could previously only be identified using tsIVs, which we show are NP-hard to find. (c): $\lambda_{xy}$ cannot be identified using tsIVs, but is identified through iterative application of Theorem 5.1 ($\lambda_{bx}$ using $a$, $\lambda_{yc}$ using $x^*$, and $\lambda_{xy}$ using $c^*$). (d): $\lambda_{x_1 y}$ is identifiable with cAV but is not captured by IC.

**Definition 2.2.** *(Weihs et al., 2017) Sets $S, T \subset V$, $|S| = |T| + 1 = k$ are a **tsIV** with respect to $\lambda_{xy}$ if (i) $De(y) \cap T = \emptyset$ (ii) The max flow between $S$ and $T' \cup \{x'\}$ in $G_{flow}$ is $k$, (iii) The max flow between $S$ and $T' \cup \{y'\}$ in $G_{flow}$ with $x' \to y', w_i' \to y'$, $\lambda_{w_i y} \in \Lambda^*$ removed is less than $k$.*

## 3 Efficiently Finding Instrumental Subsets

When identifying a structural parameter, $\lambda_{x_1 y}$, using instrumental sets, it is often the case that no instrument exists for $x_1$, but an instrumental set does exist for a subset of $y$'s parents that includes $x_1$. For example, in Fig. 1d, there does not exist any IV for $\{x_1\}$, but $\{z_1, z_2\}$ is an instrumental set for $\{x_1, x_2\}$, allowing identification of both $\lambda_{x_1 y}$ and $\lambda_{x_2 y}$. Likewise, in Fig. 2a, $\{x_1, x_2, x_5\}$ is the *only* subset of $Pa(y)$ which has a valid instrumental set ($\{z_1, z_3, z_4\}$).

One method for finding sets satisfying a criterion like IS would be to list all subsets of $y$'s incident edges, and for each subset, check if there exist corresponding variables $\{z_1, ..., z_k\}$ satisfying all requirements. This is indeed the approach that algorithms developed for the gHTC (Chen, 2016; Weihs et al., 2017) and AVS (Chen et al., 2016) take. However, enumerating all subsets is clearly exponential in the number structural parameters / edges pointing to $y$. In this section, we show that finding this parameter subset can instead be performed in polynomial-time.

First, we define the concept of "match-blocking", which generalizes the above problem to arbitrary source and sink sets in a DAG, and can be used to create algorithms for finding valid subsets applicable to IS, the gHTC, AVS, and our own identification criterion, instrumental cutsets (IC).

**Definition 3.1.** *Given a directed acyclic graph $G = (V, D)$, a set of source nodes $S$ and sink nodes $T$, the sets $S_f \subseteq S$ and $T_f \subseteq T$, with $|S_f| = |T_f| = k$, are called **match-blocked** iff for each $s_i \in S_f$, all elements of $T$ reachable from $s_i$ are in the set $T_f$, and the max flow between $S_f$ and $T_f$ is $k$ in $G$ where each vertex has capacity 1.*

To efficiently find a match-block[4], we observe that if a max flow is done from a set of variables $S$ to $T$, then any element of $T$ that has 0 flow through it cannot be part of a match-block, and therefore none of its ancestors in $S$ can be part of the match-block either:

**Theorem 3.1.** *Given a directed acyclic graph $G = (V, D)$, a set of source nodes $S$, sink nodes $T$, and a max flow $\mathcal{F}$ from $S$ to $T$ in $G$ with vertex capacity 1, if a node $t_i \in T$ has 0 flow crossing it in $\mathcal{F}$, then there do not exist subsets $S_m \subseteq S, T_m \subseteq T$ where $S_m, T_m$ are match-blocked and $t_i \in T_m$. Furthermore, for any match-block $(S_m, T_m)$, we have $|S_m \cap An(t_i)| = 0$.*

This suggests an algorithm for finding the match-block: find a max flow from $S$ to $T$, then remove elements of $T$ that did not have a flow through them, and all of their ancestors from $S$, and repeat until no new elements are removed. The procedure, given in Algorithm 1, runs in polynomial time.

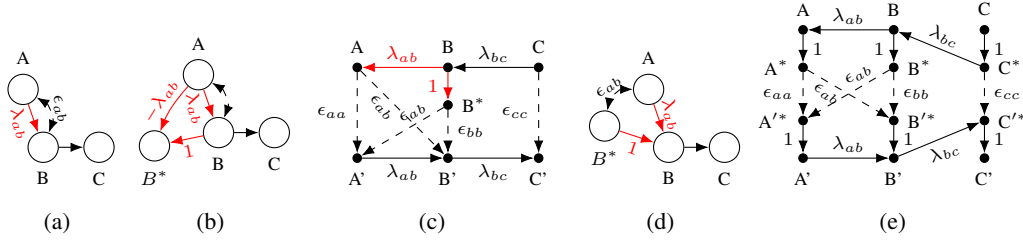

<table>
<tr><td>(a)</td><td>(b)</td><td>(c)</td><td>(d)</td><td>(e)</td></tr>
</table>

Figure 3: The graph in (a) with edge $\lambda_{ab}$ known has an auxiliary variable $B^*$ shown in (b). (c) shows a modified flow graph, which encodes the treks from $B^*$ excluding the known edge. This corresponds to a new encoding of the AV, shown in (d). Finally, (e) gives the auxiliary flow graph as described in Definition 3.2

---

**Algorithm 1** Find Maximal Match-Block given DAG $G$, source nodes $S$ and target nodes $T$

---

**function** MAXMATCHBLOCK(G,S,T)
    **do**
        $\mathcal{F} \leftarrow$ MAXFLOW$(G, S, T)$
        $T' \leftarrow \{t_i | \mathcal{F} \text{ has } 0 \text{ flow sinked by } t_i \in T\}$
        $T \leftarrow T \setminus T'$
        $S \leftarrow S \setminus An(T')$
    **while** $|T'| > 0$
    **return** $(\{s_i | s_i \in S \text{ sources a flow of } 1 \text{ in } \mathcal{F}\}, T)$
**end function**

---

The match-block can be exploited to find instrumental subsets by using $G_{flow}$ with ancestors of $y$ that don't have back-paths to siblings of $y$ as $S$ and $Pa(y)'$ as $T$. This procedure is shown to find an IS if one exists in Corollary A.3, and is implemented in Algorithm 3 of the appendix.

## 3.1 Extending IS to AVS with the Auxiliary Flow Graph

A match-block operates upon a directed graph. When using instrumental sets, one can convert the SCM to the flow graph $G_{flow}$, but the AVS algorithm exploits auxiliary variables, which are not encoded in this graph. The covariance of an auxiliary variable with another variable $y$ can be written:

$$\sigma_{b^*y} = \mathbf{E}[b^*y] = \mathbf{E}\left[\left(b - \sum_{\lambda_{x_ib} \in \Lambda^*} \lambda_{x_ib}x_i\right)y\right] = \sigma_{by} - \sum_{\lambda_{x_ib} \in \Lambda^*} \lambda_{x_ib}\sigma_{x_iy}$$

This quantity behaves like $\sigma_{by}$ with treks from $b$ starting with the known $\lambda_{x_ib}$ removed. We can therefore construct a flow graph which encodes this intuition explicitly using Definition 3.2. An example of an auxiliary flow graph can be seen in Fig. 3, which uses $b^* = b - \lambda_{ab}a$. In Fig. 3c, $B^*$ no longer has the edge $\lambda_{ab}$ to $A$, but it still has all other edges, giving it all of the same treks as $B$, except the ones subtracted out in the AV. The original variable, $B$, has an edge with weight 1 to $B^*$, making its treks identical to the standard flow graph. With this new graph, and MAXMATCHBLOCK, the algorithm for instrumental sets can easily be extended to find AVS (Algorithm 4 in appendix), which in turn is equivalent to the gHTC. We have therefore shown that both of these methods can be efficiently applied for identification, without a restriction on number of edges incoming into a node.

**Definition 3.2.** *(Auxiliary Flow Graph) Given a linear SCM $(\Lambda, \Omega)$ with causal graph $G = (V, D, B)$, and set of known structural parameters $\Lambda^*$, the auxiliary flow graph is a weighted DAG with vertices $V \cup V^* \cup V' \cup V'^*$ and edges*

    1. *$j \rightarrow i$ and $i' \rightarrow j'$ both with weight $\lambda_{ij}$ if $(i \rightarrow j) \in D$, and $\lambda_{ij} \in \Lambda^*$*

    2. *$j^* \rightarrow i$ and $i' \rightarrow j'^*$ both with weight $\lambda_{ij}$ if $(i \rightarrow j) \in D$, and $\lambda_{ij} \notin \Lambda^*$*

    3. *$i \rightarrow i^*$ and $i'^* \rightarrow i'$ with weight 1, and $i^* \rightarrow i'$ with weight $\epsilon_{ii}$ for $i \in V$*

    4. *$i^* \rightarrow j'^*$ with weight $\epsilon_{ij}$ if $(i \leftrightarrow j) \in B$*

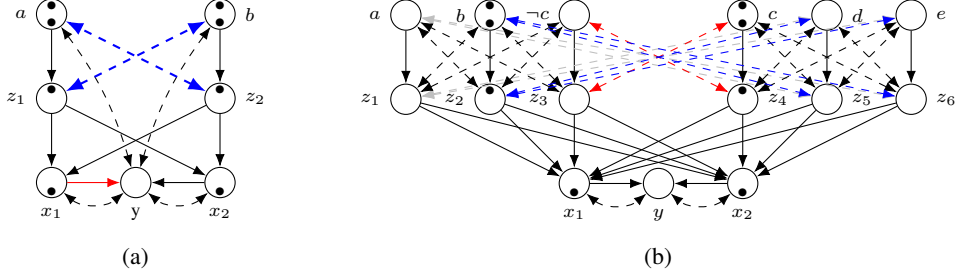

(a)                                              (b)

Figure 4: The model in (a) encodes the basic structure we exploit in our proof. The set $\{z_1, z_2\}$ is a simple conditional instrumental set only if $z_1 \leftrightarrow b$ and $z_2 \leftrightarrow a$ do not exist, since it opens the collider from $z_1$ through $b$ to $y$. (b) shows the full encoding of a 1-in-3SAT for $(a \vee b \vee \neg c) \wedge (c \vee d \vee e)$. **We have removed bidirected edges from all top nodes to $y$ in (b) for clarity**.

*This graph is referred to as $G_{aux}$. The nodes without ' are called "source", or "top" nodes, and the nodes with ' are called "sink" or "bottom" nodes. The nodes with $*$ are called "auxiliary" nodes.*

# 4    On Searching for tsIVs (Weihs et al., 2017)

There exist structural parameters that can be identified using tsIVs, which cannot be found with gHTC, AVS, nor any other efficient method. However, there currently does not exist an efficient algorithm for finding tsIVs in arbitrary DAGs. This section can be summarized with Corollary 4.1:

**Corollary 4.1.** *Given an SCM and target structural parameter $\lambda_{xy}$, determining whether there exists a tsIV which can be used to solve for $\lambda_{xy}$ in $G$ is an NP-Complete problem.*

We show this by encoding 1-in-3SAT, which is NP-Complete (Schaefer, 1978), into a graph, such that finding a tsIV is equivalent to solving for a satisfying boolean assignment.

Since tsIVs can be difficult to intuitively visualize, we will illustrate the ideas behind the proof with simple conditional instrumental sets (scIS), which we also show are NP-Complete to find (Corollary A.5). We observe a property of the graph in Fig. 4a: with $a \leftrightarrow z_2$ and $z_1 \leftrightarrow b$ removed (blue), $\{z_1, z_2\}$ can be used as scIS for $\lambda_{x_1 y}, \lambda_{x_2 y}$, since their back-paths to $y$ ($z_1 \leftarrow a \leftrightarrow y$ and $z_2 \leftarrow b \leftrightarrow y$) can be blocked by conditioning on $W = \{a, b\}$. However, if the bidirected edges are not removed, conditioning on $a$ or $b$ opens a path to $y$ using them as a collider ($z_1 \leftrightarrow b \leftrightarrow y$ and $z_2 \leftrightarrow a \leftrightarrow y$). A simple conditional instrumental set for $\lambda_{x_1 y}$ exists in this graph if and only if none of the instruments has bidirected edges to another instrument's required conditioning variable.

We exploit this property to construct the graph in Fig. 4b, which repeats the structure from Fig. 4a for each literal $l_i$ in the 1-in-3SAT formula $(a \vee b \vee \neg c) \wedge (c \vee d \vee e)$. Each clause is designed so that usage of any potential instrument, $z_i$, precludes usage of the other two potential instruments in the clause. For example, the bidirected edges in Fig. 4b from $z_3$ to $a$ and $b$ disallow usage of $z_1$ and $z_2$ as instruments once $z_3$ is used. Likewise, there are **bidirected edges** between the $c$ and $\neg c$ structures, since if $c$ is *true*, $\neg c$ cannot be *true*. Similarly, $b$ has **bidirected edges** to $z_5$ and $z_6$, since if $a$ is *true*, then $\neg c$ is false and $c$ is true, so $d$ and $e$ must be disabled. Finally, $y$ has 2 parents, $x_1, x_2$, corresponding to the two clauses. Each element of $Z$ has edges to all parents of $y$, meaning that any scIS existing in the graph must have as many instruments as there are clauses. Thus, finding an scIS for $y$ in this graph corresponds to finding a satisfying assignment for the formula. The full procedure for generating the graph, which is the same for both scIS and tsIV, is given in Theorem 4.1.

**Theorem 4.1.** *Given a boolean formula $F$ in conjunctive normal form, if a graph $G$ is constructed as follows, starting from a target node $y$:*

1. *For each clause $c_i \in F$, a node $x_i$ is added with edges $x_i \rightarrow y$ and $x_i \leftrightarrow y$*

2. *For $c_i \in F$, take each literal $l_j \in c_i$, and add nodes $z_{ij}, w_{ij}$, with edges $w_{ij} \rightarrow z_{ij}$, $w_{ij} \leftrightarrow y$, and $z_{ij} \rightarrow x_k \, \forall x_k$*

3. *For $c_i \in F$, $l_j, l_k \in c_i$ where $j \neq k$ add bidirected edge $z_{ij} \leftrightarrow w_{ik}$*

4. *$\forall c_i, c_m \in F, l_j \in c_i, l_n \in c_m$ with $i \neq m$, add a bidirected edge $z_{ij} \leftrightarrow w_{mn}$ if*

*(a)* $l_j = \neg l_n$, *or*

*(b)* $\exists l_q \in c_m$ *with* $q \neq n$ *and* $l_j = l_q$, *or*

*(c)* $\exists l_p \in c_i, l_q \in c_m$ *with* $p \neq j$ *and* $q \neq n$ *where* $l_p = \neg l_q$

*Then a tsIV exists for* $\lambda_{x_1 y}$ *in* $G$ *if and only if there is a truth assignment to the variables of* $F$ *such that there is exactly one true literal in each clause of* $F$.

## 5 The Instrumental Cutset Identification Criterion

While finding a tsIV is NP-hard, it is possible to create a new criterion, which both includes constraints from the tsIV that can be efficiently found, and exploits knowledge of previously identified edges similarly to AVS. This criterion is described in the following theorem.

**Theorem 5.1.** *(Instrumental Cutset) Let* $M = (\Lambda, \Omega)$ *be a linear SCM with associated causal graph* $G = (V, D, E)$, *a set of identified structural parameters* $\Lambda^*$, *and a target structural parameter* $\lambda_{xy}$. *Define* $G_{aux}$ *as the auxiliary flow graph for* $(G, \Lambda^*)$. *Suppose that there exist subsets* $S \subset V \cup V^*$, *with* $V^*$ *representing the set of AVs, and* $T \subseteq Pa(y^*) \setminus \{x\}$ *with* $|S| = |T| + 1 = k$ *such that*

1. *There exists a flow of size* $k$ *in* $G_{aux}$ *from* $S$ *to* $T \cup \{x\}$

2. *There does not exist a flow of size* $k$ *from* $S$ *to* $T \cup \{y\}$ *in* $G_{aux}$ *with* $x' \to y'^*$ *removed*

3. *No element of* $\{y\} \cup Sib(y)$ *has a directed path to* $s_i \in S$ *in* $G$

*then* $\lambda_{xy}$ *is generically identifiable by the equation:*

$$\lambda_{xy} = \frac{\det \Sigma_{S, T \cup \{y^*\}}}{\det \Sigma_{S, T \cup \{x\}}}$$

Instrumental Cutsets (ICs) differ from tsIVs in two fundamental ways:

1. We allow auxiliary variables, enabling exploitation of previously identified structural parameters incoming to $s_i \in S$ for identification. An example that is identifiable with ICID, but not with TSID is shown in Fig. 2c.

2. We require that $T$ is a subset of the parents of target node $y$, and that y has no half-treks to $S$. A version of IC which avoids these requirements is given in Theorem A.4 in the appendix. While this version is strictly more powerful than tsIV, finding satisfying sets can be shown to be NP-hard by a modified version of the arguments given in Section 4 (Appendix A.4.1).

$\lambda_{x_1 y}$ in Fig. 2b is an example of a parameter that can be identified using ICs. To see this, consider the paths from $\{z_1, z_2\}$ to $y$ that do not have sided intersection anywhere but at $y$. One such path set is $z_1 \to x_1 \to y$ and $z_2 \to w \to x_2 \to y$. After removing the edge $x_1 \to y$, there no longer exist 2 separate nonintersecting paths to $y$, because the node $w$ forms a bottleneck, or cut, allowing only one path to pass to $y$. According to theorem 5.1, this is sufficient to uniquely solve for $\lambda_{x_1 y}$.

In contrast, previously known efficient algorithms cannot identify $\lambda_{x_1 y}$. $z_1$, which is the only possible instrument for $x_1$, has unblockable paths to $y$ through $x_2$ and $x_3$ ($w$ cannot be conditioned, since it has a bidirected edge to $y$). Furthermore, only $z_2$ is a possible additional instrument for $x_2$ or $x_3$, giving 2 candidate instruments $\{z_1, z_2\}$ for a set of 3 parents of $y$, $\{x_1, x_2, x_3\}$, all of which need to be matched to an instrument to enable solving for $\lambda_{x_1 y}$.

In general, any coefficient that can be identified using the gHTC or AVS can also be identified using ICs. ICs, therefore, strictly subsume gHTC and AVS.

**Lemma 5.1.** *If a structural parameter* $\lambda_{xy}$ *of linear SCM* $M$ *is identifiable using the gHTC or AVS then* $\lambda_{xy}$ *is identified using IC. There also exists a model* $M'$ *such that* $\lambda_{xy}$ *is identifiable using IC, but cannot be identified using gHTC or AVS.*

Lastly, we discuss the identification power of ICs with respect to cAVs (Chen et al., 2017), which are single auxiliary conditional instruments, and can be found in polynomial-time. While there are many examples of parameters that ICs, and even the gHTC and AVS, can identify that cAVs cannot, it turns out there are also examples that the cAV can identify that ICs cannot. This is because ICs,

---
**Algorithm 2** IC solves for edges incoming to $y$ given a set of known edges $\Lambda^*$

---

**function** IC($G, y, \Lambda^*$)
    $G_{aux} \leftarrow$ AUXILIARYFLOWGRAPH($G, \Lambda^*$)
    $T \leftarrow$ all sink-node parents of $y'^*$ in $G_{aux}$
    $G_{aux}^y \leftarrow G_{aux}$ with edges $t_i \in T$ to $y'^*$ removed
    $S \leftarrow$ Source nodes in $G_{aux}^y$ which are not ancestors of $y'^*$
    $C \leftarrow$ CLOSESTMINVERTEXCUT($G_{aux}, S, T$)
    $S_f \leftarrow$ elements of $S$ that have a full flow to $C$
    $(C_m, T_m) \leftarrow$ MAXMATCHBLOCK($G_{aux}$ with edges to $C$ removed, $C, T$)
    $T_f \leftarrow$ elements of $T$ that are part of a full flow between $C \setminus C_m$ and $T \setminus T_m$
    **return** $(S_f, T_f \cup T_m, T_m)$
**end function**

---

which operate on sets of variables, do not include conditioning. In Fig. 2d, $z_1$ is a cAV for $\lambda_{x_1 y}$ when conditioned on $\{w, z_2\}$, but no IC exists, because $\lambda_{x_2 x_1}$ cannot be identified, and therefore the back path from $x_1$ through $w \leftrightarrow y$ cannot be eliminated. While a version of ICs with conditioning could be developed, the algorithmic complexity of finding parameters identifiable by such a criterion is unclear. A version of ICs with a single conditioning set would be NP-hard to find, which can be shown using a modified version of our results from Section 4 (see Appendix A.4.1). On the other hand, a version with multiple conditioning sets (one for each $s_i \in S$) would require additional algorithmic breakthroughs, due to its similarity to the as-yet unsolved gIS.

## 5.1 Efficient Algorithm for Finding Instrumental Cutsets

To demonstrate efficiency of IC, we develop a polynomial-time algorithm that finds all structural parameters identifiable through iterative application of Theorem 5.1. To do so, we show that the conditions required by Theorem 5.1 can be reduced to finding a match-block in $G_{aux}$:

**Theorem 5.2.** *Given directed graph $G = (V, D)$, a target edge $x \rightarrow y$, a set of "candidate sources" $S$, and the vertex min-cut $C$ between $S$ and $Pa(y)$ closest to $Pa(y)$, then there exist subsets $S_f \subseteq S$ and $T_f \subseteq Pa(y)$ where $|S_f| = |T_f| + 1 = k$ such that*

1. *the max-flow from $S_f$ to $T_f \cup \{x\}$ is $k$ in $G$, and*

2. *the max-flow from $S_f$ to $T_f \cup \{y\}$ in $G'$ where $x \rightarrow y$ is removed is $k - 1$*

*if and only if $x$ is part of a match-block between $C$ and $Pa(y)$ in $G$ with all edges incoming to $c_i \in C$ removed.*

Note that the "closest min-cut" $C$ required by Theorem 5.2 can be found using the Ford-Fulkerson algorithm with $Pa(y)$ as source and $S$ as sink (Picard & Queyranne, 1982).

Theorem 5.2 was proven by explicitly constructing the sets $S_f$ and $T_f$ using a match-block. The procedure for doing so is given in Algorithm 2. It works by finding a set $S_f$ which has a full flow to $C$, which in turn has a match-block to $Pa(y)$ (due to the requirement that none of the $S_f$ have paths to $y$ through $Sib(y) \leftrightarrow y$). The min-cut ensures that once $x \rightarrow y$ is removed, all paths to $y$ must still go through the set $C$, and the match-block from $C$ to $Pa(y)$ ensures that there is no way to reorder the paths to create a flow to $y$ through a different parent. This guarantees that the flow constraints are satisfied, so there is a corresponding IC. The full algorithm for finding all edges identifiable with ICs can be constructed by recursively applying the procedure on the auxiliary flow graph, as shown in Algorithm 5 (ICID)[5].

# 6 Conclusion

We have developed a new, polynomial-time algorithm for identification in linear SCMs. Previous algorithms with similar identification power had either exponential or unknown complexity, with existing implementations using exponential components. Finally, we also showed that the promising method called tsIV cannot handle arbitrarily large graphs due to its inherent computational complexity.

## Acknowledgements

Bareinboim and Kumor are supported in parts by grants from NSF IIS-1704352, IIS-1750807 (CAREER), IBM Research, and Adobe Research. Part of Chen's contributions were made while at IBM Research.

## Footnotes

[1]Such models are also referred to as structural equation models, or SEM, in the literature.

[2]Refer to Sullivant et al. (2010) and the Gessel-Viennot-Lindström Lemma (A.1) (Gessel & Viennot, 1989).

[3]Full definitions for the methods mentioned in this section are available in Appendix A.1.

[4]While there exist methods for finding solvable subsystems of equations (Duff & Reid, 1978; Sridhar et al., 1996; Gonçalves & Porto, 2016), they cannot be applied to our situation due to the requirement of nonintersecting paths in a full arbitrary DAG.

[5]A Python implementation is available at `https://github.com/dkumor/instrumental-cutsets`

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
