[Supplementary Material · supplement.pdf]

# A Appendix

This section contains the proofs of theorems mentioned in the main text, as well as efficient versions of algorithms for instrumental sets and sets of auxiliary variables. We also include a brief discussion of the difficulties of adding conditioning to instrumental cutsets.

## A.1 Definitions & Theorems from Previous Works

The proofs will make extensive use of a couple important results in the literature, which are stated here. Also given are the full definitions of certain concepts which were only briefly mentioned in the main text due to space constraints.

**Definition A.1.** *(Sullivant et al., 2010) A path $\pi$ from nodes $v$ to $w$ is a **trek** if it has no colliding arrowheads, that is, $\pi$ is of the form:*

$$v \leftarrow ... \leftarrow \leftrightarrow \rightarrow ... \rightarrow w \qquad v \leftarrow ... \leftarrow k \rightarrow ... \rightarrow w \qquad v \leftarrow ... \leftarrow w \qquad v \rightarrow ... \rightarrow w$$

**Definition A.2.** *(Foygel et al., 2012) A **half-trek** from $v$, is a trek which either starts from a bidirected edge ($v \leftrightarrow ... \rightarrow w$) or is a directed path ($v \rightarrow ... \rightarrow w$)*

**Definition A.3.** *(Sullivant et al., 2010) A **trek monomial** $\pi(\Lambda, \Omega)$ for trek $\pi$ is defined as the product of the structural parameters along the trek, multiplied by the trek's top error term covariance.*

We define $tr(v)$ and $htr(v)$ as the sets of nodes reachable from $v$ with a trek and half-trek respectively. There are two variants of trek monomial, one where $\pi$ does not take a bidirected edge, and instead has a top node $k^6$ (1), and one where the trek contains a bidirected edge $\epsilon_{ab}$ (2). We can write the covariance between $v$ and $w$, $\sigma_{vw}$, as the sum of the trek monomials of all treks between $v$ and $w$ ($\mathcal{T}_{vw}$) (3):

$$(1) \quad \pi(\Lambda, \Omega) = \epsilon_k^2 \prod_{x \rightarrow y \in \pi} \lambda_{xy} \qquad (2) \quad \pi(\Lambda, \Omega) = \epsilon_{ab} \prod_{x \rightarrow y \in \pi} \lambda_{xy} \qquad (3) \quad \sigma_{vw} = \sum_{\pi \in \mathcal{T}_{vw}} \pi(\Lambda, \Omega)$$

We can reason about the determinants of covariance matrix minors by looking at the flow graph:

**Definition A.4.** *(Sullivant et al., 2010; Foygel et al., 2012; Weihs et al., 2017) The flow graph of $G = (V, D, B)$ is the graph with vertices $V \cup V'$ containing the edges:*

- *$j \rightarrow i$ and $i' \rightarrow j'$ with weight $\lambda_{ij}$ if $i \rightarrow j \in V$*

- *$i \rightarrow i'$ with weight $\epsilon_{ii}$ for all $i \in V$*

- *$i \rightarrow j'$ with weight $\epsilon_{ij}$ if $i \leftrightarrow j \in B$*

*This graph is referred to as $G_{flow}$. The nodes without $'$ are called "source", or "top" nodes, and the nodes with $'$ are called "sink" or "bottom" nodes.*

**Lemma A.1.** *(Gessel-Viennot-Lindström Gessel & Viennot (1989); Sullivant et al. (2010); Draisma et al. (2013)) Given DAG $G = (V, E)$, with $E$ defined as a weighted adjacency matrix, and vertex sets $A, B \subseteq V$, where $|A| = |B| = l$,*

$$\det \left[ (I - E)^{-1} \right]_{A,B} = \sum_{P \in N(A,B)} (-1)^P \prod_{p \in P} e^{(p)}$$

*Here, $N(A, B)$ is the the set of all collections of nonintersecting systems of $l$ directed paths in $G$ from $A$ to $B$. $P = (p_1, ..., p_l)$ is a collection of these nonintersecting paths. $e^{(p)}$ is the product of the coefficients of $e \in E$ along path $p$, and $(-1)^P$ is the sign of the induced permutation of elements from $A$ to $B$.*

The above lemma means that a covariance matrix minor's determinant can be found by looking at nonintersecting path sets between top and bottom nodes in the flow graph (Fig. 1b). We will use $\Sigma_{S,T}$ to represent the covariance matrix minor with rows of $S$ and columns of $T$. While the flow graph has repeated edge weights, this does not affect the rank of the minor:

**Lemma A.2.** *(Weihs et al. (2017) corrolary 3.3) Let $S = \{s_1, ..., s_k\}, T = \{t_1, ..., t_m\} \subset X$, then $\Sigma_{S,T}$ has generic rank $r$ if and only if the max-flow from $s_1, ..., s_k$ to $t_1', ..., t_m'$ in $G_{flow}$ is $r$*

We can therefore say the determinant of a covariance minor from $S$ to $T$ is nonzero iff there is a full nonintersecting path set from $S$ to $T$ in the flow graph.

### A.1.1 Existing Identification Criterions

This subsection contains an unordered list of identification criteria we reference when developing our algorithms.

**Definition A.5.** *(Brito & Pearl, 2002) The set $Z$ is said to be a **generalized instrumental set** relative to $X$ and $y$ in $G$ if there exists a set $Z \subset V$ with $|Z| = |X| = k$ and set $W = \{W_{z_1}, ..., W_{z_k}\}$ with all $W_{z_i} \subset V \setminus De(y)$ such that*

    *1. There is a path set $\Pi = \{\pi_1, ..., \pi_k\}$ without sided intersection between the $Z$ and $X$,*

    *2. $W_{z_i}$ d-separates $z_i$ from $y$ in $G$ with edges $X \to y$ removed, but does not block the path from $\Pi$ starting from $z_i$*

**Theorem A.1.** *(Brito & Pearl, 2002) If there exists a generalized instrumental set to $X$, then the structural parameters $\lambda_{x_i y}$ are identifiable for $x_i \in X$.*

A simple conditional instrumental set is defined as:

**Definition A.6.** *(Van der Zander & Liskiewicz, 2016) The set $Z$ is said to be a **simple conditional instrumental set** relative to $X$ and $y$ in $G$ if there exist sets $Z \subset V$ and $W \subset V$ such that:*

    *1. There exists a set of paths $\Pi = \{\pi_1, ..., \pi_k\}$ of size $k$ without sided intersection from $Z$ to $X$*

    *2. $W$ d-separates all $Z$ from $y$ in $G$ with the edges $X \to y$ removed, but does not block any path in $\Pi$*

**Lemma A.3.** *If a simple conditional instrumental set exists for a set $X$, then all parameters $\lambda_{x_1 y}$ with $x_i \in X$ are identifiable.*

*Proof.* If there exists a simple conditional instrumental set, can directly construct a generalized instrumental set (Definition A.5), which makes the parameters identifiable by Theorem A.1. $\square$

Auxiliary variables were defined as:

**Definition A.7.** *(Chen et al., 2017) Let $M = (\Lambda, \Omega)$ be a linear SCM with variables $X$, and $\Lambda^*$ be a set of identified structural parameters. An **auxiliary variable** for $x_i \in X$ is defined as $x_i^* = x_i - \sum_{\lambda_{x_j x_i} \in \Lambda^*} \lambda_{x_j x_i} x_j$.*

An auxiliary instrumental set is defined as

**Definition A.8.** *(Chen et al., 2016) A Markovian linear SCM with graph $G$ and set of directed edges $\Lambda^*$ whose coefficient values are known is known as an auxiliary instrumental set for edges $E \subseteq Pa(y)$ if $Z^*$ is an instrumental set for $E$ where any paths through edges incoming to $z_i \in Z$ through edge $\lambda_{w z_i} \in \Lambda^*$ are considered blocked.*

The corresponding conditional auxiliary variable has the following definition:

**Definition A.9.** *(Chen et al., 2017) Given graph $G$ and set of directed edges $\Lambda^*$ whose coefficient values are known, a variable $z$ is called a conditional auxiliary instrument relative to $\lambda_{xy}$, if $z$ is a conditional instrument for $\lambda_{xy}$ in the graph with edges $w_i \to z$ and $x_i \to y$ removed where $\lambda_{w_i z}, \lambda_{x_i y} \in \Lambda^*$, and no element of conditioning set $W$ is a descendant of $z$.*

A compressed definition of tsIV was given in Definition 2.2. We include the full version here for completeness:

**Theorem A.2.** *(Weihs et al., 2017) Let $G = (V, D, B)$ be a mixed graph, $w_0 \to v \in G$, and suppose that the edges $w_1 \to v, ..., w_l \to v \in G$ are known to be generically (rationally) identifiable. Let $G_{flow}^*$ be $G_{flow}$ with the edges $w_0' \to v', ..., w_l' \to v'$ removed. Suppose there are sets $S \subset V$ and $T \subset V \setminus \{v, w_0\}$ such that $|S| = |T| + 1 = k$ and*

1. $De(v) \cap (T \cup \{v\}) = \emptyset$,

2. the max-flow from $S$ to $T' \cup \{w'_0\}$ in $G_{flow}$ equals $k$, and

3. the max-flow from $S$ to $T' \cup \{v'\}$ in $G^*_{flow}$ is $< k$,

then $w_0 \rightarrow v$ is rationally identifiable by the equation

$$\lambda_{w_0 v} = \frac{\left|\Sigma_{S, T \cup \{v\}}\right| - \sum_{i=1}^{l} \left|\Sigma_{S, T \cup \{w_i\}}\right|}{\left|\Sigma_{S, T \cup \{w_0\}}\right|}$$

## A.2 Match-Blocking

The matchblock definition is restated here for convenience:

**Definition 3.1.** *Given a directed acyclic graph $G = (V, D)$, a set of source nodes $S$ and sink nodes $T$, the sets $S_f \subseteq S$ and $T_f \subseteq T$, with $|S_f| = |T_f| = k$, are called **match-blocked** iff for each $s_i \in S_f$, all elements of $T$ reachable from $s_i$ are in the set $T_f$, and the max flow between $S_f$ and $T_f$ is $k$ in $G$ where each vertex has capacity 1.*

The algorithm for finding a match-block (Algorithm 1) relies on the following theorem:

**Theorem 3.1.** *Given a directed acyclic graph $G = (V, D)$, a set of source nodes $S$, sink nodes $T$, and a max flow $\mathcal{F}$ from $S$ to $T$ in $G$ with vertex capacity 1, if a node $t_i \in T$ has 0 flow crossing it in $\mathcal{F}$, then there do not exist subsets $S_m \subseteq S, T_m \subseteq T$ where $S_m, T_m$ are match-blocked and $t_i \in T_m$. Furthermore, for any match-block $(S_m, T_m)$, we have $|S_m \cap An(t_i)| = 0$.*

*Proof.* Suppose not. That is, suppose that $\exists t_i \in T$ such that a max-flow $\mathcal{F}$ gives 0 flow through it, yet there exist subsets $S_m \subseteq S, T_m \subseteq T$ with $t_i \in T_m$, such that $De(S_m) \cap T \subseteq T_m$, and there is a full flow $F_m$ from $S_m$ to $T_m$ in the graph where each vertex has capacity 1 ($|S_m| = |T_m| = |\mathcal{F}_m|$).

We reason about the intersection of the two flows $\mathcal{F}$ and $\mathcal{F}_m$. Our goal is to show that we can modify $\mathcal{F}$ to include the paths of $\mathcal{F}_m$, without intersecting paths in $\mathcal{F}$ to $T \setminus T_m$, thus creating a new flow larger than $\mathcal{F}$ - which is a contradiction, since $\mathcal{F}$ is a max-flow.

Suppose that $\mathcal{F}$ has non-zero flow from $S_f \subseteq S$ to $T_f \subset T$. For any $t_j \in T_m \cap T_f$ we can simply remove the original path in $\mathcal{F}$ to $t_j$, resulting in a new flow $\mathcal{F}'$, which is of size $|\mathcal{F}| - |T_m \cap T_f|$.

Next, we show that the flow paths in $\mathcal{F}_m$ cannot intersect with any flow paths from $\mathcal{F}'$. This is because $De(S_m) \cap T \subseteq T_m$ requires that all matching descendants of elements in $S_m$ are in $T_m$, meaning that any flow-path $p_k \in \mathcal{F}'$ intersecting flow-path $p'_l \in F_m$ must have an endpoint on an element in $T_m$ - but all such paths were removed in $\mathcal{F}'$.

We combine the paths of $\mathcal{F}'$ and $\mathcal{F}_m$, giving a new flow $\mathcal{F}''$ from $S$ to $T$. Finally, since $t_i \notin T_f$, $|T_m \cap T_f| < |T_m|$, so:

$$|\mathcal{F}| - |T_m \cap T_f| + |T_m| = |\mathcal{F}''|$$
$$|\mathcal{F}| < |\mathcal{F}''|$$

A contradiction ($\mathcal{F}$ is a max-flow). There cannot be any satisfied subset $T_m$ containing $t_i$.

Finally, no ancestor of $t_i$ can be part of a satisfied subset, since they all have paths to $t_i$, which completes the proof. $\square$

**Corollary A.1.** *Given directed acyclic graph $G = (V, D)$ and sets of source nodes $S, T$, their maximal match-blocked subsets $S_m, T_m$ can be found in polynomial time.*

*Proof.* See Algorithm 1. At each iteration, at least one node is eliminated from the feasibility set, meaning that given $n$ nodes, the algorithm runs at most $n$ max-flow queries, each of which is computable in polynomial time. $\square$

**Corollary A.2.** *Suppose that $S_m, T_m$ constitute a match-block. The match-block $S'_m, T'_m$ found using Algorithm 1 is such that $S_m \subseteq S'_m$ and $T_m \subseteq T'_m$.*

*Proof.* At each step of the algorithm, only nodes $t_i, ...t_j$ that have 0 flow crossing through them are removed from $T$. These nodes cannot be part of $T_m$ by Theorem 5.2. Similarly, only ancestors to $t_i, ..., t_j$ are removed from $S$, which likewise cannot be part of $S_m$. Therefore, no elements of $S_m$ nor $T_m$ were removed by the algorithm, meaning that $S_m \subseteq S'_m$ and $T_m \subseteq T'_m$. $\qquad\square$

**Corollary A.3.** *Given mixed graph $G = (V, D, B)$, there exists a valid instrumental subset $S_m \subseteq V \setminus De(Sib(y))$ to $T_m \subseteq Pa(y)$ if and only if $S_m, T_m$ is a match-block between $V \setminus De(Sib(y))$ and $Pa(y)$ in $G_{flow}$.*

*Proof.* $\Rightarrow$: Given a valid instrumental subset, it is also a match-block, because there are an equal number of instruments and parents of $y$, and there exists a system of nonintersecting paths from $S_m$ to $T_m$. Furthermore, if any of the $S_m$ has paths to $Pa(y) \setminus T_m$, then they are dependent on $y$ in the graph with the edges $T_m \rightarrow y$ removed, which means that they are not an IS. Finally, the IS can't have elements of $De(Sib(y))$, since it would mean that $y$ is not-d-separated from the instruments. As such, the sets satisfy the requirements of a match-block.

$\Leftarrow$: Suppose there is a valid match-block, then there is a valid instrumental set. The match-block guarantees the requirements of an IS directly. The restriction of $S$ to elements without back-paths to $y$ forces all paths to $y$ to go through $Pa(y)$.

The corresponding algorithm is given in Algorithm 3 $\qquad\square$

---

**Algorithm 3** Find Maximal Instrumental Subsets given graph $G$, target variable $y$

---

    **function** MAXIS(G,y)
        $G_{flow} \leftarrow$ FLOWGRAPH$(G)$
        $Z \leftarrow (An(y, G_{flow}) \cap$ SOURCENODES$(G_{flow})) \setminus$ SOURCENODESOF$(De(Sib(y)))$
        $(Z_f, X_f) \leftarrow$ MAXMATCHBLOCK$(G_{flow}, Z, Pa(y)')$
        **return** $(Z_f, X_f)$
    **end function**

---

### A.2.1 Auxiliary Flow Graph

This section develops results that show the auxiliary flow graph can be used instead of $G_{flow}$, and that it encodes treks through auxiliary variables. For reference, $G_{aux}$ is defined as:

**Definition 3.2.** *(Auxiliary Flow Graph) Given a linear SCM $(\Lambda, \Omega)$ with causal graph $G = (V, D, B)$, and set of known structural parameters $\Lambda^*$, the auxiliary flow graph is a weighted DAG with vertices $V \cup V^* \cup V' \cup V'^*$ and edges*

    *1. $j \rightarrow i$ and $i' \rightarrow j'$ both with weight $\lambda_{ij}$ if $(i \rightarrow j) \in D$, and $\lambda_{ij} \in \Lambda^*$*

    *2. $j^* \rightarrow i$ and $i' \rightarrow j'^*$ both with weight $\lambda_{ij}$ if $(i \rightarrow j) \in D$, and $\lambda_{ij} \notin \Lambda^*$*

    *3. $i \rightarrow i^*$ and $i'^* \rightarrow i'$ with weight 1, and $i^* \rightarrow i'$ with weight $\epsilon_{ii}$ for $i \in V$*

    *4. $i^* \rightarrow j'^*$ with weight $\epsilon_{ij}$ if $(i \leftrightarrow j) \in B$*

*This graph is referred to as $G_{aux}$. The nodes without $'$ are called "source", or "top" nodes, and the nodes with $'$ are called "sink" or "bottom" nodes. The nodes with $*$ are called "auxiliary" nodes.*

**Lemma A.4.** *Given a linear SCM $(\Lambda, \Omega)$ with causal graph $G = (V, D, B)$, a set of known structural parameters $\Lambda^*$, and defining $V^* = \{v_1^*, ..., v_k^*\}$ as*

$$v_i^* = v_i - \sum_{\lambda_{v_j v_i} \in \Lambda^*} \lambda_{v_j v_i} v_j$$

*the sum over each path of products of weights along the path from $s \in V \cup V^*$ to $t \in V'$ in the auxiliary flow graph $G_{aux}$ encodes the covariance $\sigma_{st}$.*

*Proof.* We already know that $G_{flow}$ encodes treks in the graph (Sullivant et al., 2010). If $s \in V$, we notice that the sum of paths can be constructed by combining the paths from $s$ that are not passing $s^*$, and the paths from $s^*$ multiplied by 1 - which results in the treks, identically to $G_{flow}$.

If $s^* \in V^*$, we notice that the set of treks from $s^*$ to a variable $y$ can be seen as a subset of the treks from $s$

$$\sigma_{sy} = \Big(\text{treks not starting from any } \lambda_{a_j s} \in \Lambda^*\Big) + \Big(\text{treks starting from one of the } \lambda_{a_j s} \in \Lambda^*\Big)$$

$$\sigma_{sy} = \Big(\text{treks not starting from any } \lambda_{a_j s} \in \Lambda^*\Big) + \sum_j \lambda_{a_j s}\sigma_{a_j y}$$

$$\sigma_{sy} - \sum_j \lambda_{a_j s_i}\sigma_{a_j y} = \Big(\text{treks not starting from any } \lambda_{a_j s} \in \Lambda^*\Big)$$

$$\sigma_{s^* y} = \Big(\text{treks not starting from any } \lambda_{a_j s} \in \Lambda^*\Big)$$

Using this result, we can conclude that the covariance of $s_i^*$ with any variable behaves as if the edges from $s$ to $a_j$ in $G_{flow}$ did not exist, but all other paths were identical to $G_{flow}$. This is exactly the construction given in $G_{aux}$. $\qquad\square$

Next, we show directly that the Gessel-Viennot-Lindrström Lemma still holds for the auxiliary graph. While the statement is lengthy, it simply states that we can just use the nonintersecting path sets in the new graph to determine values of determinants of minors of the covariance matrix where each variable also has an "auxiliary" version of itself, where known effects are removed.

**Lemma A.5.** *(Auxiliary Gessel-Viennot-Lindström) Given a linear SCM $(\Lambda, \Omega)$ with causal graph $G = (V, D, B)$, a set of known structural parameters $\Lambda^*$, and defining $V^* = \{v_1^*, ..., v_k^*\}$ as $v_i^* = v_i - \sum_{\lambda_{v_j v_i} \in \Lambda^*} \lambda_{v_j v_i} v_j$, for any vertex sets $A \subseteq V \cup V^*$ and $B \subseteq V$, where $|A| = |B| = l$,*

$$\det \Sigma_{A,B} = \sum_{P \in N(A,B)} (-1)^P \prod_{p \in P} e^{(p)}$$

*Here, $N(A, B)$ is the the set of all collections of nonintersecting systems of $l$ directed paths in the auxiliary flow graph $G_{aux}$ from $A$ to $B$. $P = (p_1, ..., p_l)$ is a collection of these nonintersecting paths. $e^{(p)}$ is the product of the weights along path $p$, and $(-1)^P$ is the sign of the induced permutation of elements from $A$ to $B$.*

*Proof.* This is a direct consequence of Lemma A.1 and Lemma A.4. $\qquad\square$

**Corollary A.4.** *Auxiliary Instrumental Sets can be found in polynomial time*

*Proof.* The proof is identical to Corollaries A.1 and A.3, with the only difference being that the auxiliary flow graph is used in place of $G_{flow}$. The full algorithm is shown in Algorithm 4. $\qquad\square$

---

**Algorithm 4** Finds all edges identifiable using AVS in polynomial time

---

**function** AVS(G,y,$\Lambda^*$)
    $G_{aux} \leftarrow$ AUXILIARYFLOWGRAPH$(G, \Lambda^*)$
    $T \leftarrow$ all sink-node parents of $y'^*$ in $G_{aux}$
    $G_{aux}^y \leftarrow G_{aux}$ with edges $t_i \in T$ to $y'^*$ removed
    $S \leftarrow$ Source nodes in $G_{aux}^y$ which are not ancestors of $y'^*$
    **return** MAXMATCHBLOCK$(G_{aux}, S, T)$
**end function**
**function** AVSID(G)
    $\Lambda^* \leftarrow \emptyset$
    **do**
        **for all** $y \in G$ **do**
            $(\_, T_m) \leftarrow$ AVS$(G, y, \Lambda^*)$
            $\Lambda^* \leftarrow \Lambda^* \cup \{\lambda_{ty}|t \in T_m\}$
        **end for**
    **while** at least one parameter was identified in this iteration
    **return** $\Lambda^*$
**end function**

---

### A.3 NP-Hardness of tsIV and scIV

We first relate scIV to tsIV, then we construct a supporting lemma used within our main NP-hardness proof, and finally, we prove that tsIVs and scIVs are NP-Hard, and therefore NP-Complete.

**Theorem A.3.** *If there exists a simple conditional instrumental set usable to solve for $\lambda_{xy}$, then there exists a tsIV that can be used to solve for $\lambda_{xy}$*

*Proof.* Suppose there is a simple instrumental set described by $Z, X, W$, where $W$ is the conditioning, $X$ is a set of $y$'s parents, and $Z$ is the set of instruments. We claim that the corresponding tsIV has $S = Z \cup W$ and $T = W \cup X \setminus \{x\}$ for any $x \in X$.

To witness, observe that condition 1 of Theorem A.2 is satisfied, since $X$ and $W$ are non-descendants of $y$, and condition 2 is also satisfied, since we can construct a full flow between $S$ and $T \cup \{x\}$ by adding a path from each $w_i$ to $w_i'$, and using the paths $\Pi$ from Definition A.6 from $Z$ to $X$. If a path from $z$ to $x$ crosses a collider $w_i$, we construct the path $z$ to $w_i'$ and $w_i$ continuing on the path to $x'$.

Finally, we can focus on condition 3 of Theorem A.2. We will prove it holds by contradiction. Suppose that there exists a full flow from $S$ to $T \cup \{y\}$ in the flow graph with edge $x' \to y'$ removed.

We know $W$ d-separates $Z$ from $y$ when the edges $\lambda_{x_iy}, x_i \in X$ are removed. Since $T$ contains all $x_i \in X$ except $x$ itself, none of the edges $x_i' \to y'$ can be taken, since their corresponding vertex $x_i'$ is already part of a path. Likewise, since $x' \to y'$ is removed, none of the paths can take that edge either.

Nevertheless, the $S$ must have a valid matching to $T \cup \{y\}$ for a full flow to exist. We now show that this is impossible by induction.

Let $s_0 \in S$ be the element matched to $y'$ in the full flow. Either $s_0 \in Z$ or $s_0 \in W$.

We know that $s_1 \notin Z$, since by d-separation, all treks from $z_i$ to $y'$ that don't pass the removed edges are intersected by elements of $W$, which corresponds to blocking both the top and bottom nodes of the flow graph. This means that $s_1 \in W$.

Suppose that for $s_1, ..., s_i$, each $s_k \in W$. Let $s_{i+1} \in S$ be matched to $s_i'$ in the full flow. Suppose $s_{i+1} \in Z$. Then it means that there is a path $s_{i+1}, s_i, ..., s_1, y$ across v-structures to $y$, making the element of $z$ not d-separated from $y$, a contradiction. Therefore $s_{i+1}$ must be in $W$. But there is a finite number of $W$, meaning that the only possible elements to match to $s_n'$ after all $W$ are already matched to something will be elements of $Z$, a contradiction.

This proof showed that any full matching of $S$ to $T \cup \{y\}$ has a confounding path across v-structures, which means that $z_i$ was not d-separated from $y$, a contradiction. $\square$

**Lemma A.6.** *Given sets $S$ and $T$, and a full flow $\mathcal{F}_x$ of size $k$ between $S$ and $T \cup \{x\}$, if $\mathcal{F}_x$ has an $s_i \in S$ matched to $x$, then if the bidirected edge $s_i \leftrightarrow y$ exists, there exists a flow of size $k$ between $S$ and $T \cup \{y\}$, and there does not exist a tsIV for $\lambda_{xy}$.*

*Proof.* We can construct a flow $\mathcal{F}_y$ of size $k$ by replacing the path $s_i$ to $x$ with $s_i \leftrightarrow y$, which is a flow of size $k$ from $S$ to $T \cup \{y\}$, and means that no tsIV exists.

$\square$

**Theorem 4.1.** *Given a boolean formula $F$ in conjunctive normal form, if a graph $G$ is constructed as follows, starting from a target node $y$:*

1. *For each clause $c_i \in F$, a node $x_i$ is added with edges $x_i \to y$ and $x_i \leftrightarrow y$*

2. *For $c_i \in F$, take each literal $l_j \in c_i$, and add nodes $z_{ij}, w_{ij}$, with edges $w_{ij} \to z_{ij}$, $w_{ij} \leftrightarrow y$, and $z_{ij} \to x_k \ \forall x_k$*

3. *For $c_i \in F$, $l_j, l_k \in c_i$ where $j \neq k$ add bidirected edge $z_{ij} \leftrightarrow w_{ik}$*

4. *$\forall c_i, c_m \in F, l_j \in c_i, l_n \in c_m$ with $i \neq m$, add a bidirected edge $z_{ij} \leftrightarrow w_{mn}$ if*

   (a) *$l_j = \neg l_n$, or*
   (b) *$\exists l_q \in c_m$ with $q \neq n$ and $l_j = l_q$, or*

(c) $\exists l_p \in c_i, l_q \in c_m$ with $p \neq j$ and $q \neq n$ where $l_p = \neg l_q$

Then a tsIV exists for $\lambda_{x_1 y}$ in $G$ if and only if there is a truth assignment to the variables of $F$ such that there is exactly one true literal in each clause of $F$.

*Proof.* First, we convert the formula into a version without repeated literals in any clause by removing variables of repeated literals from all clauses where they appear (a repeated literal must be false, since otherwise the clause would have 2 trues, and would also force the remaining literal to be true). Similarly, we simplify out formulae with a literal and its negation in a single clause. We can simplify the formula such that each clause does not have any repeated statements. We operate upon this converted formula.

Consider $F$ being made up of $k$ clauses $c_1, ..., c_k$, where clause $i$ has literals $l_{i1}, l_{i2}, l_{i3}$.

$\Rightarrow$: **We prove that there exists a tsIV in $G$ if $F$ is satisfiable**, by constructing a simple conditional instrumental set (Definition A.6) for the $X$ variables, and using Theorem A.3 to show that a corresponding tsIV exists.

Let $l_{is} \in c_i$ be the true literal of a satisfying assignment. We construct $W = \bigcup_{c_i \in F} \{w_{is}\}$ and $Z = \bigcup_{c_i \in F} \{z_{is}\}$. We now show that these sets describe a simple conditional instrumental set.

Let $G'$ be the graph where all edges $X \to y$ are removed. Each $z_i \in Z$ has the correspoding $w_i$ conditioned. None of the bidirected edges between $z$ and $w$ are between elements of $Z, W$, since that would mean that the assignments are incompatible (either two literals true in a single clause, or a literal in a different clause being incompatible with the assignment implied by $l_{is}$). This means that each $z_i \in Z$ is d-separated from $y$ in $G'$, and therefore $Z, W, X$ is a simple instrumental set (Figs. 6g and 6h).

$\Leftarrow$: **We show that if there does not exist a valid assignment to $F$, then there do not exist sets $S$ and $T$ that can be used as a tsIV for $\lambda_{x_1 y}$.** We will exploit conditions 2 and 3 of Theorem A.2 to show that whenever there is a flow of size $k$ between $S$ and $T \cup \{x_1\}$, then there is also a flow of size $k$ between $S$ and $T \cup \{y\}$ in the graph with $x_1' \to y'$ removed.

This is easiest to prove through contradiction. Suppose that there exists a valid tsIV despite there being no satisfying 1-in-3SAT assignment to $F$. This means that there exists a flow $\mathcal{F}_x$ of size $k$ between $S$ and $T \cup \{x_1\}$ but no flow of size $k$ between $S$ and $T \cup \{y\}$.

Since flows correspond to nonintersecting path sets, then $\mathcal{F}_x$ represents a matching of nonintersecting paths, one of which is from some $s_i \in S$ to $x_1$. We know that this $s_i$ cannot be any of the $X$ or $W$ nodes, since they have bidirected edges to $y$ in $G$ (Lemma A.6).

This means that the only possibility for generating a valid tsIV is for $s_i \in Z$ to match with $x_1$ in $\mathcal{F}_x$.

We now know that any valid tsIV has an element $s_i = z_i \in Z$ matched with $x_1$. Notice that $z_i$ has paths to $y$ through all of the $x_2', ..., x_n'$. We could construct a flow $\mathcal{F}_y$ e.g. with the path $z_i \to z_i' \to x_1'$ replaced with a path $z_i \to z_i' \to x_j' \to y$, which once again corresponds to a full flow, meaning that no such tsIV exists (see Figs. 7a and 7b). Since $z_i$ is matched with $x_1$, in must have an unblocked path to $x_1$ through one of the $z_j'$, so all $x_i'$ must be in $T$ to disallow constructing such a $\mathcal{F}_y$.

Next, we require that each of the $x_i \in X \setminus \{x_1\}$ added to $T$ has a corresponding matched variable in $S$. None of the $X$ can be used for this function, since they have bidirected edges to $y$, which could be used to construct a full flow to $y$ as shown in Figs. 7c and 7d.

Likewise, none of the $W$ can match to these $X$, because, once again, if there was an $x_j'$ matched to a $w_l$, we could construct a new flow of size $k$, $\mathcal{F}_y$, which has $z_i$ matched with $x_j'$, and uses the $w_l \in W$'s bidirected edge to $y$ to create a full flow.

This means that we must have at least $k$ elements of $Z$ in $S$. Each of these $z_j \in S \cap Z$ which is matched to elements of $X$ has an open path to $x_1$, meaning that all of them need to have $w_j \in S$ to block the back-path from $z_j$ that could match with $y$ through $w$'s bidirected edge.

At this point, we have shown that any tsIV in a graph constructed as given in Theorem 4.1 must have in $S$ a set of $k$ $Z$ variables, called $Z_k$ which all have unblocked paths to the $X$, and a set $W_k = W \cap Pa(Z_k)$, meaning $Z_k \cup W_k \subseteq S$ and $X \setminus \{x\} \subset T$.

Next, we need to add nodes that match to $W_k$ to $T$, since the flow must be of full size from $S$ to $T \cup \{x\}$.

We first claim that the corresponding nodes cannot be elements of $Z'_k$. Suppose not, that is, suppose that $\exists w_i \in W_k$ matched to $z'_i$. The only possible way for this to be true is for $z_i$'s path to $x_i$ to be as shown in Figs. 6e and 6f. However, any such matching can be flipped so that it is $w_i$ matching to $x'_j$ and $z_i$ to $z'_i$, which would mean the existence of a full flow, and therefore no tsIV (Lemma A.6).

Furthermore, building upon this result, we will claim that any $z_i$ matched with $x_j$ must have a valid matching through $z_i \to z'_i \to x'_j$. That is, we claim that $z'_i$ cannot be blocked. This can be seen by contradiction. Suppose $z'_i \in T$. Then there must be a $w_j$, $z_j$, or even $x_j$ which matches with $z'_i$ ($i \neq j$). However, we can then take the flow $\mathcal{F}_x$, and match $z_i$ with $z'_i$, and have the node originally matched with $z'_i$ take the bidirected edge $w_j \leftrightarrow y$ instead. This means we can construct a full flow $\mathcal{F}_y$, meaning that a tsIV cannot exist. We can therefore assume that all $k$ matchings from the $z_i$ to $x_j$ go through the nodes $z'_i$ instead of taking bidirected edges (if $\mathcal{F}_x$ has a flow from $z_i$ through a bidirected edge to $x_j$, can replace it with the flow through $z'_i$ to create another full flow $\mathcal{F}'_x$).

**Recap:** We know that the $k$ nodes that match with the $k$ $x$ nodes must all be $z$ nodes, and each of those nodes $z_i$ must have its associated $w_i \in S$ to block the possible path to $y$ (Fig. 5a). Finally, the $w_i$ cannot be matched to $z'_i$, and $z_i$ must have its matching path be $z_i \to z'_i \to x'_j$.

**Claim:** We now claim that if a tsIV exists, with sets $S$ and $T$, then it must have a full flow where all $k$ $z$ nodes matching to the $x$ have their corresponding $w_i$ matched to $w'_i$, meaning that the $w_i$ does not match through a bidirected edge, but rather matches to itself. That is, we claim that there exists a flow that has $k$ matching substructures of the form $z_i \to z'_i \to x'_i$ and $w_i \to w'_i$ (Figs. 6c and 6d). We will call these "active literals".

We will prove that there must be $k$ active literals in this flow by contradiction. Suppose not. That is, suppose that the maximal number of active literals is $m < k$. Choose the flow $\mathcal{F}$ with all $m$ active literals. Next, choose one of the remaining $z_j$ that is matched to $x_j$, which is not part of an active literal. $z_j$ must have $w_j \in S$, because the path in Fig. 5b must be blocked. The $w_j$ must be matched to an element other than $w'_j$, since it would be an active literal, a contradiction. The matched edge must be a descendant of $w_i$ in $G_{flow}$, meaning that the only candidate is $z_j$ with $j \neq i$ ($x_j$ was already matched to the $k$ $z_i$ values (Fig. 5c), and $z_i$ was already shown to be impossible).

We know $z_j$ is not an active literal, so we have the paths shown in Fig. 5d, which means that $w'_j$ must be blocked by adding $w'_j$ as a sink (otherwise we could create a full flow $\mathcal{F}_y$). We now observe the source matched to $w'_j$ in $\mathcal{F}_x$. $w_j$ cannot be the source (Fig. 5e), and same is true of $z_j$, due to the the same reason: $w_j \leftrightarrow y$ gives a way to match to $y$ (Fig. 5f). Finally, once again, no value of $x$ can be part of the matching, due to its bidirected edge to $y$ (Fig. 5g)

This means that the only possible matching is with another $z_m$ connected to $w_j$ with bidirected edge, which would then need to have $w_m$ blocked (Fig. 5h), and can't be an active literal, since those are already matched to $x$. Now, we have returned to the situation in Fig. 5a, where the options are using $w_m$ to match, or having a match through a bidirected edge.

The matches in $\mathcal{F}$ can continue through further bidirected edges, but due to the finite amount of literals in the full formula, at some point the chain will end with a $w_n$ matched to $w'_n$. We can then reorder this matching to create a new active literal (Fig. 5i). This means that we have constructed a matching that will have the same total flow as $\mathcal{F}$, but has 1 more active literal, which is a contradiction, since $\mathcal{F}$ already has the maximal number of active literals.

We therefore conclude that there must be $k$ active literals in the tsIV.

**Recap:** We know that any tsIV has a full flow $\mathcal{F}$ which includes $k$ active literals, meaning that there are $k$ substructures where $z_i \in Z_k$ has $z_i \to z'_i \to x_j$ and $w_i \to w'_i$ are in $\mathcal{F}$

We exploit the knowledge that there is no satisfying 1-in-3SAT assignment to show that there must exist at least two of the $z_i, z_j \in Z_k$ and corresponding $w_i, w_j \in W_k$ such that there are bidirected edges $w_i \leftrightarrow z_j$ and $w_j \leftrightarrow z_i$, allowing us to construct a full flow including $y$ as shown in Figs. 7e and 7f.

With this, we have ensured that there is no possible tsIV if there is no satisfying truth assignment where exactly one literal is true in each clause.

$\square$

**Corollary 4.1.** *Given an SCM and target structural parameter $\lambda_{xy}$, determining whether there exists a tsIV which can be used to solve for $\lambda_{xy}$ in G is an NP-Complete problem.*

*Proof.* 1-in-3SAT was shown to be NP-complete by Schaefer (1978). By Theorem 4.1, we can solve 1-in-3SAT with an algorithm for tsIV. The corresponding graph is computable from the boolean formula in polynomial time. Likewise, Theorem A.2 describes a polynomial-time procedure for determining whether a given set can be used as a tsIV. Therefore, the problem is NP-complete. $\square$

**Corollary A.5.** *Given an acyclic DAG G and edge $\lambda_{xy}$, finding a simple conditional instrumental set which can be used to solve for $\lambda_{xy}$ in G is an NP-Complete problem.*

*Proof.* 1-in-3SAT was shown to be NP-complete by Schaefer (1978). The proof of Theorem 4.1, constructed a simple conditional instrumental set for $X$ whenever there was a satisfying assignment. The corresponding graph is computable from the boolean formula in polynomial time. Likewise, (Van der Zander & Liskiewicz, 2016) describes a polynomial-time procedure for determining whether a given set can be used as a simple instrumental set. Finally, since a tsIV does not exist whenever there is no satisfying assignment in Theorem 4.1, and whenever a simple conditional isntrumental set exists, a corresponding tsIV also exists (Theorem A.3), no simple instrumental set exists if there is no satisfying assignment. Therefore, finding simple conditional instruments is an NP-complete problem. $\square$

Figure 5: Graphs used to demonstrate elements of the proof of Theorem 4.1

## A.4 Instrumental Cutsets

**Theorem 5.1.** *(Instrumental Cutset) Let $M = (\Lambda, \Omega)$ be a linear SCM with associated causal graph $G = (V, D, E)$, a set of identified structural parameters $\Lambda^*$, and a target structural parameter $\lambda_{xy}$. Define $G_{aux}$ as the auxiliary flow graph for $(G, \Lambda^*)$. Suppose that there exist subsets $S \subset V \cup V^*$, with $V^*$ representing the set of AVs, and $T \subseteq Pa(y^*) \setminus \{x\}$ with $|S| = |T| - 1 = k$ such that*

1. *There exists a flow of size $k$ in $G_{aux}$ from $S$ to $T \cup \{x\}$*

2. *There does not exist a flow of size $k$ from $S$ to $T \cup \{y\}$ in $G_{aux}$ with $x' \to y'^*$ removed*

3. *No element of $\{y\} \cup Sib(y)$ has a directed path to $s_i \in S$ in $G$*

*then $\lambda_{xy}$ is generically identifiable by the equation:*

$$\lambda_{xy} = \frac{\det \Sigma_{S, T \cup \{y^*\}}}{\det \Sigma_{S, T \cup \{x\}}}$$

*Proof.* Our proof mirrors the work of Weihs et al. (2017), except with the auxiliary flow graph instead of $G_{flow}$. We construct the auxiliary flow graph $G_{aux}$. Condition 1 guarantees that $G_{aux}$ has a set of nonintersecting paths from $S^*$ to $T' \cup \{x'\}$, which means that $\Sigma_{s^*, T \cup \{x\}}$ is full rank (Lemma A.5).

Figure 6: Graphs used to illustrate elements of the proof of Theorem 4.1

(a)                    (b)

(c)                    (d)

(e)                    (f)

Figure 7: Graphs used to illustrate elements of the proof of Theorem 4.1

Then, condition 2 and 3 guarantees that there is no set of nonintersecting paths from $S^*$ to $T' \cup \{y'\}$ that does not go through the edge $x'$. Each set of paths from $S^*$ to $T' \cup \{x'\}$ can be extended to a path from $S^*$ to $T' \cup \{y'\}$ by adding $x' \to y'$ to the path that ends at $x'$. Using Lemma A.5 we have

$$\det \Sigma_{S^*, T \cup \{y\}} = \lambda_{xy} \det \Sigma_{S^*, T \cup \{x\}}$$

which gives an equation for $\lambda_{xy}$ after division.

If edge from $w_i$ incoming to $y$ is known, then all treks to $y$ through $\lambda_{w_i y}$ can be observed with $\lambda_{w_i y} \Sigma_{S^*, T \cup \{x\}}$ (once again, due directly to the interpretation of determinants of minors in the covariance matrix as sets of nonintersecting paths due to Lemma A.5), giving all paths not passing any of the known edges with:

$$\det \Sigma_{S^*, T \cup \{y\}} - \sum_{\lambda_{w_i y} \in \Lambda^*} \det \Sigma_{S^*, T \cup \{w_i\}}$$

This leads to the full statement of the formula, mirroring the original equation of Theorem A.2.

$\square$

**Lemma 5.1.** *If a structural parameter $\lambda_{xy}$ of linear SCM $M$ is identifiable using the gHTC or AVS then $\lambda_{xy}$ is identified using IC. There also exists a model $M'$ such that $\lambda_{xy}$ is identifiable using IC, but cannot be identified using gHTC or AVS.*

*Proof.* First, neither gHTC nor AVS can identify $\lambda_{x_1 y}$ in Fig. 2b, but it is identifiable with IC.

Let $S$ be a set of auxiliary variables that satisfies the requirements of an AVS to a set of edges $X$ where $X \subseteq Pa(y)$.

Then there is a full flow between $S^*$ and $X$ (there are paths with no sided intersection), and since all $Pa(y)$ reachable from $S^*$ are in $X$, and $|X| = |S|$, they correspond to a match-block, meaning that the minimal cutset will also have a matchblock, so by Theorem 5.2, it satisfies the rules of Theorem 5.1.

Finally, AVS subsumes gHTC, so ICs subsume both. $\square$

**Theorem A.4.** *(Auxiliary tsIV) Let $G = (V, D, B)$ be a mixed graph, $w_0 \to v \in G$, and suppose that the edges $w_1 \to v, ..., w_l \to v \in G$ are known to be generically (rationally) identifiable. Let $G'_{aux}$ be $G_{aux}$ with the edges $w'_0 \to v', ..., w'_l \to v'$ removed. Suppose there are sets $S \subset V$ and $T \subset V \setminus \{v, w_0\}$ such that $|S| = |T| + 1 = k$ and*

1. *$De(v) \cap (T \cup \{v\}) = \emptyset$,*

2. *the max-flow from $S$ to $T' \cup \{w'_0\}$ in $G'_{aux}$ equals $k$, and*

3. *the max-flow from $S$ to $T' \cup \{v'\}$ in $G'_{aux}$ is $< k$,*

*then $w_0 \to v$ is rationally identifiable by the equation*

$$\lambda_{w_0 v} = \frac{\left| \Sigma_{S, T \cup \{v\}} \right| - \sum_{i=1}^{l} \left| \Sigma_{S, T \cup \{w_i\}} \right|}{\left| \Sigma_{S, T \cup \{w_0\}} \right|}$$

*Proof.* Identical to Theorem 5.1. $\square$

### A.4.1 A Discussion of Conditonal IC Complexity

Many models previously only identifiable with conditional cAVs are identifiable with IC, such as the one in Fig. 8a. The key here is that in many situations, the back paths can be removed through edges identifiable in previous iterations of the algorithm, and there is no need to block paths to parents of $y$, since the IC can simply use all parents of $y$ in the set.

This shows why the proof in Theorem 4.1 cannot be directly applied to IC. The structure shown in Fig. 4a has its edge $\lambda_{a z_1}$ and $\lambda_{b z_2}$ identifiable directly, and therefore there is no need for conditioning at all - the edge is identifiable with standard IC.

Figure 8: In (a), $z$ a conditional IV for $\lambda_{x_1 y}$ given $w_1$ and $w_2$. However, we can use $w_1$ to solve for $\lambda_{w_1 z}$ and $\lambda_{z w_2}$, at which point we can identify $\lambda_{x_1 y}$ using Theorem 5.1. In (b), we show the example of an IC for $\lambda_{x_2 y}$ where $S = \{z_2, z_4\}$ and $T = \{x_1\}$. Critically, the back-path through the bidirected edge to $y$ from $z_2$ is blocked by the path from $z_4$ to $x_1$. The naïve construction of a half-trek IC would have $S' = \{z_1, z_2\}$, which would consequently allow a path from $z_2 \leftarrow z_3 \leftrightarrow y$. In (c) is demonstrated the reason a match-block is required even when the closest cutset is limited to parents of $y$. $z_1$ has closest cutset at $x_1'$, but does not have a match-block, since $x_1'$ also has a path to $x_2'$, which does not have a corresponding match. Finally, (d) demonstrates why incoming edges to the cut $C$ are removed, with the example where an IC exists for $\lambda_{x_1 y}$ but $x_2$ and $x_3$ both being descendants of $x_1$ if edges incident to $w$ are not removed.

However, we can use the structure in Fig. 2d, which cannot have its back paths removed to replace the original structure of a literal in the proof. The construction would be identical to the one given in Theorem 4.1, but with 2 $x$ variables per clause, and each literal consisting of the structure in Fig. 2d. That way, we can show that both the auxiliary tsIV and a conditional version of IC would also be NP-hard to find.

### A.4.2 Algorithm for Instrumental Cutsets

Before developing an algorithm for IC, we first show some helper lemmas which allow us to gain an intuition about the problem.

**Lemma A.7.** *Given a DAG $G = (V, D)$, a set of sources $S$, and sinks $T$, and a vertex min-cut $C$ between $S$ and $T$ closest to $T$, then if there exists a match-block between $C_m \subseteq C$ and $T_m \subseteq T$, then $C_m = T_m$.*

*Proof.* Suppose that $\exists c_i \in C_m \setminus T$. We can then replace $C_m$ in $C$ with $T_m$, to create a closer min-cut of the same size as $C$ (a contradiction), because $|C_m| = |T_m|$ and $T_m$ blocks all paths outgoing from $C_m$ to $T$. Furthermore, if $\exists c_i \in C_m \setminus T_m$ where $c_i \in T$, then it is not part of a match-block, since it has an unblocked path to $c_i \in T$ - another contradiction. □

This means that a match-block between $C$ and $T$ can be described with a single set $C_m \subseteq T$, which describes both start and endpoints of the matchblock. Note, however, that we still need to find a full match-block, as shown in the example in Fig. 8c, where even when $T = Pa(y)$, no match-block exists, because $x_1$ has a path to $x_2$.

**Lemma A.8.** *Given a DAG $G$, a set of sources $S$, sinks $T$, and a vertex min-cut $C$ between $S$ and $T$, there exist max flow $\mathcal{F}_C$ from $S$ to $T$ which can be decomposed into max-flows $\mathcal{F}_S$ from $S$ to $C$ and $\mathcal{F}_T$ from $C$ to $T$, with $|\mathcal{F}_S| = |\mathcal{F}_T| = |\mathcal{F}_C| = |C|$.*

*Proof.* This is by definition of a vertex min-cut. □

**Lemma A.9.** *Given a DAG $G$, a set of sources $S$, sinks $T$, if there is a vertex max-flow $\mathcal{F}$ of size $k$ between $S$ and $T$, with nonzero-flow start and endpoints $S_f \subseteq S$, $T_f \subseteq T$ respectively, and $S_m \subseteq S_f, T_m \subseteq T$ constitute a match-block, then $\mathcal{F}$ can be decomposed into $\mathcal{F}^-$ between $S^- = S_f \setminus S_m$ and $T^- = T_f \setminus T_m$ of size $k - |S_m|$, and a full flow $\mathcal{F}_m$ between $S_m$ and $T_m$.*

*Proof.* We know that $\mathcal{F}_m$ is a flow of size $|S_m| = |T_m|$. By the definition of match-block, we know that all descendants of $S_m$ that are in $T$ are also in $T_m$. This means that any path starting from $S_f$ that crosses the descendants of $S_m$ must end on $T_m$. Suppose that $\exists s_i \in S^-$ that is matched with $t_i \in T_m$. Consequently, one of the $S_m$ cannot be matched, a contradiction, since $S_m$ *all* have nonzero flow in $\mathcal{F}$. We can therefore conclude that $\mathcal{F}$ can be decomposed into a full flow from $S_m$ to $T_m$, contained entirely in the descendants of $S_m$ ($\mathcal{F}_m$), and a flow $\mathcal{F}^-$ which does not cross the descendants of $S_m$, from $S^-$ to $T^-$. Since the flows can be decomposed into a flow through the match-block, and a flow avoiding the match-block, we know that $|\mathcal{F}| = |\mathcal{F}_m| + |\mathcal{F}^-|$, which completes the proof (remember $|\mathcal{F}_m| = |S_m|$ and $|\mathcal{F}| = k$).

$\square$

**Lemma A.10.** *Given a DAG $G$, a set of source nodes $S$, sinks $T$, and a node $x \in An(T), x \notin De(S)$, if there does not exist a max-flow $\mathcal{F}$ from $S$ to $T$, and a path $p$ from $x$ to any $t_i \in T$ such that $p$ doesn't cross any flow from $\mathcal{F}$, then there exists a closest min-cut $C$ between $S$ and $T$ where at least one element $c_i \in C$, such that $c_i \notin S$, or all paths from $x$ cross $S$.*

*Proof.* Suppose that the max-flow $\mathcal{F}$ is of size $k$. We now take a new max-flow $\mathcal{F}'$ from $S \cup \{x\}$ to $T$. If this flow were of size $k + 1$, we could decompose it into a flow of size $k$ between $S$ and $T$, and a flow of size 1 between $x$ and $T$ (since we already know the max-flow between $S$ and $T$ is $k$). We can construct a path $p$ that contradicts the theorem's statement using the flow from $x$ to $T$. This means that $\mathcal{F}'$ must be of size $k$.

We therefore have $|S \cup \{x\}| > k$ and $|T| > k$, meaning that there is a bottleneck of size $k$ between the two sets (ie, a set of nodes forming a min-cut). All paths from $x$ to $T$ must cross elements of a closest min-cut $C$.

If there exists a path from $x$ to $T$ not crossing $S$, the closest-min-cut of the path must be in $De(S)$, since otherwise the full min-cut would be $k + 1$, giving a flow of $k + 1$, a contradiction.

We therefore know that there exists a cut $C$ of size $k$ contained in $De(S)$, which cuts $S \cup \{x\}$ from $T$. The same cut therefore must cut $S$ from $T$ - and if $x$ has a path to $T$ not crossing $S$, an element of $C$ must be along the path, meaning that $c_i \in C$, but $c_i \notin S$. $\square$

**Theorem 5.2.** *Given directed graph $G = (V, D)$, a target edge $x \to y$, a set of "candidate sources" $S$, and the vertex min-cut $C$ between $S$ and $Pa(y)$ closest to $Pa(y)$, then there exist subsets $S_f \subseteq S$ and $T_f \subseteq Pa(y)$ where $|S_f| = |T_f| + 1 = k$ such that*

1. *the max-flow from $S_f$ to $T_f \cup \{x\}$ is $k$ in $G$, and*

2. *the max-flow from $S_f$ to $T_f \cup \{y\}$ in $G'$ where $x \to y$ is removed is $k - 1$*

*if and only if $x$ is part of a match-block between $C$ and $Pa(y)$ in $G$ with all edges incoming to $c_i \in C$ removed.*

*Proof.* $\Rightarrow$: Suppose that a match-block $C_m = T_m$ exists between $C$ and $Pa(y)$ (see Lemma A.7) when incident edges to $C$ are removed. We show that we can construct sets $S_f, T_f$ satisfying conditions 1 and 2 for all $x \in T_m$. Let $\mathcal{F}$ be a (vertex) max-flow from $S$ to $Pa(y)$ in the graph with all edges present. Next, let $S_f \subseteq S, T_f \subseteq Pa(y) \setminus \{x\}$ be the nodes of $S$ and $T$ respectively with nonzero flow in $\mathcal{F}$. We define $k = |S_f|$, automatically satisfying condition 1. We can decompose $\mathcal{F}$ into a max-flow of size $k = |C|$ from $S_f$ to $C$ ($\mathcal{F}_S$) and from $C$ to $T_f \cup \{x\}$ ($\mathcal{F}_T$) using Lemma A.8. Finally, we define $C_m^- = C \setminus C_m$ and $T_m^- = T_f \setminus C_m$.

With $x \to y$ removed and $x \in T_m$ removed from target nodes, we have one of the $c_i \in C_m$ not matched with any element of $T_m \cup \{y\}$, by definition of match-block over $T_m$, making the maximum flow between $C_m$ and $T_m \cup \{y\} \setminus \{x\}$ be $|C_m| - 1$ (Lemma A.9). There remains a full flow between $C_m^-$ and $T_m^-$, so all of the other $C$ elements have flow through them, making a path from $c_i$ not able to pass over $c_j \neq c_i$. This makes the full max-flow from $C$ to $T_f$ be $k - 1$ by combining the match-blocked paths and non-match-blocked paths. Finally, all paths from $S_f$ to $y$ must cross $C$, so we have satisfied condition 2.

$\Leftarrow$: We now show that if a match-block between $C$ and $T_f \cup \{x\}$ does not exist containing $x$, then either condition 1 or 2 is violated. We can find sets $S_f, T_f$ satisfying condition 1 (otherwise $S_f$ has

no path to $x$, and no match-block exists to $x$). The question, then, is whether any such sets also satisfy condition 2.

Given any candidate set $S_f, T_f$ satisfying condition 1, we have the closest min-cut $C$ between $S_f$ and $T_f \cup \{x\}$. Suppose for the sake of contradiction that condition 2 is also satisfied. That is, all flows through $C$ to $T_f \cup \{x\}$ must pass through $x \to y$. This means that $x$ is part of the closest min-cut, ie, $x \in C$. By the theorem's conditions, all edges incoming to $c_i \in C$ were removed, so there are no edges incoming to $x$. If $x$ has no path to any $t_i \in T_f$, then $x$ is match-blocked with itself (ie, can use $S_f = \{x\}, T_f = \emptyset$), a contradiction. We know that the flow between $C \setminus \{x\}$ and $T_f$ must be $k-1$ (if it were $k$, we would not have condition 2 satisfied). If there exists a flow between $C \setminus \{x\}$ and $T_f$ that does not block a path from $x$ to some $t_i \in T_f$, then condition 2 is satisfied by appending this path to the flow, constructing a new flow of size $k$, a contradiction. Finally, in the case when all paths from $x$ to $T_f$ are blocked by all flows between $C \setminus \{x\}$ and $T_f$ we invoke Lemma A.10 to claim that there exists a closer min-cut than $C$, a contradiction.

$\square$

---

**Algorithm 5** ICID recursively applies IC to all nodes

---

    **function** ICID(G)
        $\Lambda^* \leftarrow \emptyset$
        **do**
            **for all** $y \in G$ **do**
                $(\_, \_, T_m) \leftarrow \text{IC}(G, y, \Lambda^*)$
                $\Lambda^* \leftarrow \Lambda^* \cup \{\lambda_{ty} | t \in T_m\}$
            **end for**
        **while** at least one parameter was identified in this iteration
        **return** $\Lambda^*$
    **end function**

---

## Footnotes

[6]Note also that we can have a trek from $v$ to $v$, including a trek that takes no edges at all, which would be simply $\epsilon_{vv}$