[Reviews · NeurIPS 2019]

Reviewer 1



The paper seems to address an important computational problem in identifying particular generalizations of the IVs, proposing an algorithm that can be implemented in polynomial time. The importance of this algorithm depends on how useful these generalizations are. The authors mention that they can be applied to arbitrary structural causal models (as opposed to linear SCMs that are commonly assumed in the literature). As these references are relatively new and somewhat specialized, it would be useful for the authors to include a more detailed introduction to these methods, highlighting the main ideas. For example, there is a big gap from the second to the third paragraph in Section 2; also in Definition 2.2. Another concern is lack of numerical studies to illustrate the importance of the proposed method. Maybe a run time comparison would better illustrate the benefits of the proposed method in realistic settings.

Reviewer 2



As noted above, I find this work to be quite interesting and also quite densely informative. I suspect that, should this paper be accepted, the authors are planning to further extend this work into a longer form. In that case, I would definitely suggest expanding on the examples and the introduction, since this was a challenging read for someone who is not intimately familiar with all of the previous work. In the future, I would also appreciate the implementation of their suggested algorithm and some simulation comparisons of their work and other existing work. Of course, the worst case scenario is clear from their theoretical results, but I would still be interested to see how these algorithms compare in a simulation. Minor comments: - lines 31 and 32, I believe the indexes should be reversed \lambda_ij = 0, that is, whenever x_j is not a direct cause of x_i - the references sometimes use initials and sometimes full first names. - line 365 corollary -> Corollary

Reviewer 3



UPDATE: Thank you for the thoughtful response, those changes should improve the things that were unclear to me. There is a rich recent literature on identification criteria for linear structural causal models, but most of the recently proposed criteria largely ignore the question of efficient computability. This paper answers important questions in this area by given efficient algorithms for some criteria, while showing others to be NP-complete. The paper is original and generally clear and of high quality. Minor comments: l100: double "a" l104: the equation you refer to is in the supplement, which should be mentioned here. There should also be a general mention in the main paper informing readers about the supplement. Figure 2a: some labels are right between two nodes, making it hard to see which they belong to. Definition 3.1: "for each s_i ... are in the set T_j" is extremely hard to parse. I suggest: "any path from s_i \in S_f to t \in T \setminus T_f has an intermediate node in S_F \cup T" l149: You didn't mention yet what "this problem" is; I believe it is finding a maximal match-blocking? (Where "maximal" is inclusionwise?) Theorem 3.1: Also hard to read, in particular the precedence of the "... and ... or ..." at the end is ambiguous. Maybe the "there exists ... and ..." could be rewritten to "for all ... such that ..., we have ..."? Algorithm 1: After the while loop, T will contain only nodes that participate in the flow, while S may not (e.g. for the graph s_1 -> t <- s_2). So the line after the while loop should modify S, not T. (BTW, these match-blocks seem related to the Edmond-Gallai decomposition, though I believe that is defined for undirected graphs. You might want to check if you can find literature for a more efficient algorithm, or theoretical results. For example, I would guess that all maximal match-blocks have the same T, and so as a corollary, all maximal match-blocks are maximum.) Supplement l445: period missing

[Author Response · NeurIPS 2019]

# 1  All Reviewers

We thank the reviewers for their input and time. We were happy to see that our contribution was considered informative and original. We will respond to each reviewer's comments individually.

# 2  Reviewer 1

We certainly agree with your assessment of the importance of computationally efficient methods for identification. As you noticed, there is a large body of specialized previous work that we build upon, which makes a relatively limited introduction/preliminaries section unavoidable given the restricted space. To partly remedy the issue, we will add the relevant definitions and theorems from the literature into the supplemental material, and include a note referencing them in the preliminaries. We will also revise and improve the preliminaries to alleviate your concerns about the conceptual jump between our treatment of structural parameters and flow graphs.

Our reference to "arbitrary SCM" in the abstract was meant to imply arbitrary graphical structures. We recognize that this might have caused confusion, so we have changed it to read "arbitrary linear SCM". We do assume linearity in our work and focus on algorithmic aspects of identification methods for linear models.

Finally, given the space constraints, and the level of technical specialization in our references, we have opted to dedicate more space towards explaining our main theoretical results rather than adding an experimental section. Still, we do plan to add experiments in a longer report following from this paper.

# 3  Reviewer 2

We were delighted to hear that you found the paper interesting and deserving of publication, much appreciated. We certainly recognize that our paper compressed quite a bit of details, especially with regard to the previous literature. To help interested readers gain context, we have opted to copy the relevant definitions and theorems from previous works into the supplement, and to revise the preliminaries for greater clarity. The paper will not fit a full explanation of the previous works, but we expect (and hope) that these changes will make our work easier to parse.

While we think that space restrictions make adding an experimental section impractical, we plan to release an implementation of the algorithms in the paper (in Python), including the IC and AVS algorithms. We plan to make the link available in the text whenever anonymity is no longer an issue.

Finally, we thank you for the corrections and suggestions in the "Minor Comments".

# 4  Reviewer 3

We are happy that you found our paper of high quality and are very grateful for the detailed corrections. We have noticed the typo in Algorithm 1, and have updated all algorithms in the paper to more directly mirror our Python code, which we will release with the paper. We have also reworded the theorems to improve legibility and precision. In particular, Theorem 3.1 now reads:

 *Given a directed acyclic graph $G = (V, D)$, a set of source nodes $S$, sink nodes $T$, and a max flow $\mathcal{F}$ from $S$ to $T$ in $G$ with vertex capacity 1, if a node $t_i \in T$ has 0 flow crossing it in $\mathcal{F}$, then there do not exist subsets $S_m \subseteq S, T_m \subseteq T$ where $S_m, T_m$ are match-blocked and $t_i \in T_m$. Furthermore, for any match-block $(S_m, T_m)$, we have $|S_m \cap An(t_i)| = 0$.*

It is also correct that all max-match-blocks have the same T – we will add a more explicit proof of this fact into the supplement. Finally, as was noticed, the Edmond-Gallai decomposition is defined on undirected graphs, and is focused on single edges. Our task requires paths in a DAG, and these paths are between two separate sets of nodes. Still, we would be glad to use previous literature to improve our match-block algorithm even further. The relevant works we found are listed in footnote 2 on page 4, but cannot be directly applied due to complexities arising from the full DAG.

Other minor corrections: Thanks, fixed!

[Meta-Review · NeurIPS 2019]

The paper proposes a method to efficiently find instrumental subsets for identification in linear acyclic SCMs. The reviewers think that the method is interesting and relevant. An improvement to its evaluation would be the addition of an experimental section -- the authors indicated that they will add it in the revised version of the paper.